# Active Dual Line-Laser Scanning for Depth Imaging of Piled Agricultural Commodities for Itemized Processing Lines

**DOI:** 10.3390/s24082385

**Published:** 2024-04-09

**Authors:** Mohamed Amr Ali, Dongyi Wang, Yang Tao

**Affiliations:** 1Fischell Department of Bioengineering, University of Maryland, College Park, MD 20742, USA; mali93@terpmail.umd.edu; 2Department of Biological and Agricultural Engineering, University of Arkansas, Fayetteville, AR 72701, USA; dongyiw@uark.edu

**Keywords:** 3D optical sensor, laser triangulation, active line scanner, depth imaging, surface profilometry, surface topography

## Abstract

The accurate depth imaging of piled products provides essential perception for the automated selection of individual objects that require itemized food processing, such as fish, crabs, or fruit. Traditional depth imaging techniques, such as Time-of-Flight and stereoscopy, lack the necessary depth resolution for imaging small items, such as food commodities. Although structured light methods such as laser triangulation have high depth resolution, they depend on conveyor motion for depth scanning. This manuscript introduces an active dual line-laser scanning system for depth imaging static piled items, such as a pile of crabs on a table, eliminating the need for conveyor motion to generate high-resolution 3D images. This advancement benefits robotic perception for loading individual items from a pile for itemized food processing. Leveraging a unique geometrical configuration and laser redundancy, the dual-laser strategy overcomes occlusions while reconstructing a large field of view (FOV) from a long working distance. We achieved a depth reconstruction MSE of 0.3 mm and an STD of 0.5 mm on a symmetrical pyramid stage. The proposed system demonstrates that laser scanners can produce depth maps of complex items, such as piled Chesapeake Blue Crab and White Button mushrooms. This technology enables 3D perception for automated processing lines and offers broad applicability for quality inspection, sorting, and handling of piled products.

## 1. Introduction

With the advent of advanced imaging sensors and machine learning, machine vision systems have become integral to automating tasks on industrial lines [1]. Robotic pick-and-place operations have found wide application in food processing and manufacturing industries [2] to alleviate labor shortages [3]. Vision-guided industrial processes are well-suited for managing uniform, manufactured objects with predefined dimensions. However, handling piles of non-uniform objects produced by nature, such as seafood, produce, and other agricultural products, presents significant challenges that require intelligent sensing and recognition. Agricultural products such as fruits and vegetables can be separated or isolated using shaker/vibratory conveyors [4,5]. However, other products, such as crustaceans, often get entangled, complicating the separation process for itemized processing. Therefore, sensing methods that involve 3D imaging and perception to capture the object geometry and morphology are essential to achieve autonomous robotic tasks.

Depth imaging has emerged as a significant advancement in smart food processing [6]. Yet, current depth sensors have limitations and offer insufficient depth resolution for accurately capturing small agricultural commodities where millimeter precision is required. Furthermore, current optical depth imaging methods do not fully resolve the field of view due to optical occlusions around pile apices.

Three broad classes of 3D imaging principles exist: interferometry, Time-of-Flight (TOF), and optical triangulation [6]. Interferometry techniques such as optical coherent tomography have the highest depth resolution and accuracy in the order of micrometers, making them successful in the medical field. However, they are scarce in industrial settings because of their limited depth range, small field of view, and slow scan speed [7].

TOF-based sensors utilize the time and phase differences between emission and reflected light to estimate the distance between the object and the sensor. They are more suitable candidates for large FOVs and depth ranges, which is why LiDARs, one type of TOF-based depth imaging modality, predominate the autonomous driving field [8]. However, TOF sensors, such as Intel Real Sense L515, have low depth reconstruction accuracy, as shown in Figure 1, making them unsuitable for industrial lines where products are small and require feature recognition in millimeter resolutions. To achieve a 1 mm depth resolution, a TOF sensor requires timing a pulse that only lasts 6.6 picoseconds, which cannot be attained in silicone at room temperature [9]. Recent advances in Direct-TOF sensors and superconductivity enable techniques such as Single-Photon LiDARs to achieve submillimeter resolution at a high frame rate. However, these sensors operate at large working distances (on the order of hundreds of meters), limiting their use in indoor industrial plants. Single-Photon LiDARs are also susceptible to fluctuating operating temperatures and electrical noise [10], which is prevalent in industrial settings. Additionally, these systems are costly and have a large footprint because they necessitate substantial cooling equipment.

Optical triangulation methods balance system complexity, accuracy, and depth range. Optical triangulation uses geometrical optics to describe the relationship between the cameras and/or structural light to measure depth. Passive optical triangulation methods, such as stereo-vision and digital photogrammetry, reconstruct three dimensions using multiple camera views. However, this process requires finding corresponding pixels from separate camera frames. Passive acquisition methods are limited in industrial applications because they struggle to reconstruct accurate depth maps if they cannot find the correspondence [11]. Low disparity estimates can arise when one of the cameras is obstructed or does not have the same line of sight as the other camera, failing to reconstruct depth maps. Similarly, low disparity estimates can also arise if the target objects have repetitive regions or surfaces with low texture [12].

To remedy the shortcomings of passive acquisition, active optical triangulations utilize illumination sources to offer unique feature points, making it feasible to calculate depth one point at a time. Essentially, active acquisition methods fix the correspondence issue of passive counterparts by introducing artificial features. However, the resulting depth image quality relies on the illumination method. Illumination methods such as dot laser scanners utilize a MEMS mirror to move a circular dot laser in the X-Y plane of the FOV. In this method, the vertical scanning speed is usually the system’s bottleneck, leading to longer acquisition times [13]. Two-dimensional structured light dramatically speeds up scanning time, but the illumination patterns limit the lateral resolution of the depth map. Multiple illumination pattern design strategies utilize color, shape, and frequencies to generate varying image features [14,15]. In practice, the complexity of the pattern (usually fringes or pseudo-random light-coded dots) and the ambiguity resulting from the pattern designs are a pair of contradictions. Encoding complex 2D patterns minimizes the reconstruction ambiguities and leads to higher accuracy, but decoding the complex patterns becomes computationally expensive and more fallible. Environmental interferences such as vibrations affect the decoding of patterns on an image-wide scale, making them unreliable for industrial lines [16].

Line-scan-based optical triangulation methods can achieve relatively high depth map accuracy while avoiding the potential crosstalk between image feature patterns that 2D structured lighting suffers. Consequently, they received significant attention in industrial applications [17]. The system typically comprises a fixed illumination laser and a sensing camera setup. An optical encoder is mounted on the conveyor shaft, triggering camera image acquisition. However, this synchronization strategy requires the conveyor to move to achieve the scanning. The conveyor motion approach is limited when objects in a static pile, such as a pile of crabs, need to be scanned and selected for itemized processing. If multiple-depth images are required to complete the itemized picking task, the conveyor needs to move back and forth several times to create the scans. The items are susceptible to changing position during the conveyor movement and, thus, lose their image registration position.

Furthermore, depth line scan cameras suffer from occlusion, where high objects obstruct illumination or the optical path to the nearby regions (Figure 2). Robot [18] or gantry-based [19] laser illumination could theoretically circumvent optical laser obstruction and movement in product piles, but these methods are expensive and slow. The need for extensive mechanical movement of large systems constrains the scanning speed. In contrast, our approach demonstrated that manipulating the laser light path with a galvanometer achieves much faster scanning speeds for static objects [20]. This is the first attempt in the literature to address the obstruction issue using fast dual active line-laser scanning.

This manuscript aims to develop an active laser scanning system to produce depth images for agricultural processing lines. The imaging system has an overhead configuration to effectively capture large fields of views from a considerable working distance while ensuring high reconstruction performance and depth range for the bulk imaging of piles. The system is designed to accommodate diverse products with various textures, shapes, and colors and function effectively in the mechanically and optically noisy conditions typical of industrial environments.

## 2. Materials and Methods

The 3D imaging system employs a pair of line lasers (with different colors) and galvanometers to manipulate light paths for scanning piled objects on industrial lines. Featuring an overhead camera configuration, each laser galvanometric unit is strategically placed upstream or downstream from the camera. This design offers multiple advantages over single laser line scanners and line scan cameras. First, it enables scanning on static industrial lines, which is crucial for maintaining the position and orientation of piled products. Second, the overhead camera setup is ideal for a bird’s-eye view, aiding robotic tasks in expansive workspaces. Third, unlike passive 3D reconstructions using a stereoscopic configuration, our system effectively scans textured and textureless products, expanding its utility for diverse industrial lines. Lastly, the dual-laser redundancy in our setup overcomes the limitations of traditional galvanometric methods, especially in illuminating obstructed areas where objects’ apices peak in the middle of the field of view (FOV) and block the light path. The laser redundancy also extends the height range in image regions where tall objects might cause laser shifts to fall outside the FOV. The proposed system yields two depth maps that enhance our depth reconstruction performance when merged.

### 2.1. Dual Line Laser Active Scanning Machine Vision System

As shown in Figure 3, the 3D scanning system is comprised of an overhead CMOS camera (Basler AG, Ahrensburg, Germany, acA2000-340kc) with a focusable lens (Fujinon, Tokyo, Japan, HF16A-2) and two galvanometric units up- and downstream (Figure 3). The camera is connected to a frame grabber (Matrox Imaging, Dorval, QC, Canada, Rapixo CL Pro) via two Camera Link connectors. The frame grabber and camera communicate via the GenICam protocol, which operates on Low-Voltage Differential Signaling (LVDS)—an electromagnetically proof standard for industrial lines with high electromagnetic noise from motors and other equipment. The frame grabber and camera capture images at 1030 × 1086 pixels, corresponding to a FOV of approximately 350 × 371 mm. The lasers’ thickness is 1.4 mm (measured at the tabletop in the middle of the FOV). Therefore, the FOV can be scanned in 250 frames. The camera is configured to ten taps (1X10-1Y) at 75 MHz clock speed. Both the camera’s Auto White Balance and Auto-Gain are turned off to minimize inter-frame lighting variability and reduce acquisition time. The exposure time is 2 milliseconds to reduce background noise and maintain the laser’s high-amplitude signal. This configuration enables an acquisition rate of 360 frames/sec.

Each galvanometric unit consists of a focusable 20 mW line laser (CivilLaser 650 nm and 532 nm) placed in front of a silver-coated mirror (Figure 3). The laser colors match the optical bands where the CMOS sensor exhibits optimal quantum efficiency. The mirror is attached to a single-axis galvanometer (Thorlabs, Newton, NJ, USA, GVS011) and reflects the light downward through an opening in the galvanometric unit assembly. Both galvanometer actuators are powered through a dedicated power supply (Thorlabs GPS011-US) and controlled through an analog, high-precision motor driver. The position of the rotatory mirror is determined by an analog input voltage signal with a resolution of 0.5 V per degree.

The camera and galvanometric units are mounted on an overhead aluminum rail. Several aspects of the system need to be aligned for optimal performance. First, a digital leveler aligns the tabletop and overhead rails to ensure their planes are parallel to the ground. Second, the laser position and orientation are manually adjusted inside the galvanometric unit to enable the line illumination to run horizontally across the frame at the same pixel height (ν pixels). The horizontal alignment is checked by comparing the line laser’s position in the rightmost and leftmost pixels. Third, both line lasers are focused on the middle of the image frame to obtain optimal focusing throughout the FOV. It is important to note that laser thickness slightly varies due to laser defocusing at the edges of FOV. The defocusing occurs due to the change of incidence angles and varying travel distances. Finally, the input voltage controlling the mirror’s rotational position (τ) is empirically adjusted to illuminate the first and last row of the image frame to determine the lasers’ range. These input voltages correspond to the maximum and minimum projection laser angles. All other intermediate laser projection positions and their respective motor voltages are computed based on linear interpolation of the relevant range. This linear interpolation essentially determines the scan step size of the motor.

A multifunctional I/O board (ACCESS I/O Inc., San Diego, CA, USA, PCIe-DA16-6) commands an alternating sequence of hardware triggers to the frame grabber for image capture and step scan motor movement. The scanning routine begins by capturing a background color image where the lasers are absent in the FOV, followed by 250 laser-scan images while the lasers sweep across the FOV. The acquisition procedure utilizes static settings to minimize pixel-level variation between two subsequent frames. As shown in Figure 4, the background image is subtracted from all laser images, resulting in the laser signature only. This procedure reduces the effect of environmental lighting variations [13]. RGB and HSV color space thresholds are applied on laser images to split the laser-containing images into two stacks of binary masks (250 green and 250 red masks). Segmented pixels are clustered and narrowed down to one pixel based on a column-wise mean-shift algorithm [21].

Column-wise subtraction between baseline and shifted positions facilitates geometric calculations to reconstruct the depth map. Each set of masks reconstructs a depth map independently. If both lasers reconstruct a particular pixel, the algorithm averages the findings of the depth height. If neither laser reconstructs a pixel, missing regions are filled by nearby heights. Matrox Imaging Library and C++ imaging SDKs are adopted in post-processing and depth map reconstruction. The details and reconstruction procedure are elaborated in the following sections.

### 2.2. Optical Triangulation and Object Height Estimation

The presented depth imaging design is based on optical triangulation. Figure 5 illustrates a side-view diagram of the geometrical theory behind optical triangulation. When an object is placed in a laser line’s path, the overhead camera observes a shift from the baseline measurement (Y = Y_wb_) to a new object-related measurement (Y = Y_wo_). Here, Y_w_ denotes the y-axis of the world coordinate frame, and the subscripts b and o correspond to baseline and object-related measurements, respectively. Using a line laser, its illumination traverses the X-axis (across the image frame) at a given Y-axis location. Assuming the angle between the line laser and the ground at Z_w_ = 0 is denoted as θ (where 0 ≤ θ ≤ π and θ ≠ $\frac{\pi }{2}$), the relationship between the laser position shift and object height could be computed using the following Equation:(1)$$Z_{wo}=\left(y_{wb}-y_{wo}\right)\;\times \;\tan\left(\theta \right)$$
where Y_wb_ and Y_wo_ are computed from the camera pixel values represented by ($u_{b}$, $v_{b}$) and ($u_{o}$, $v_{o}$), where u and v are the pixel locations in the rows and columns of the image frame. The laser angle θ can also be inferred from the motor/mirror angle.

### 2.3. Camera-to-World Calibration and Lens Distortion Correction

Two essential types of calibrations are conducted to obtain precise 3D reconstruction from a conventional camera. Firstly, image-to-world frame calibration accounts for lens distortions and scaling based on the camera pinhole model. Secondly, the laser-to-camera calibration establishes the relationship between shifted laser positions and corresponding known heights to form the basis of the trigonometric model. The laser-to-camera calibration relies on calculation from the primary image-to-world coordinate calibration. The model equations related to the camera-to-world calibration are described as follows.

Cameras’ optics project a 3D world onto a 2D image plane through extrinsic and intrinsic transformations. The extrinsic parameters describe the six-dimension position and orientation of the camera frame relative to the world coordinate reference point. The intrinsic parameters of the lens maps position from the camera frame to pixel locations on the camera sensor. Multiple chessboard images are experimentally utilized with an out-of-shelf iterative linear regression algorithm [22] to obtain the camera’s intrinsic parameters. The origin of the world coordinate system (X_w_, Y_w_, Z_w_) is marked on the food-safe HDPE table as a reference position to obtain the extrinsic matrix.

As mentioned in Section 2.1, laser-scanning images are processed to convert pixel positions from the image frame to the camera frame and finally to world coordinates in millimeters. This is the reverse order of transformations from the intrinsic and extrinsic parameters. In the following procedure, the inverse of intrinsic and extrinsic parameters is performed to transform the images from image frame to camera frame and then to world frame.

To transform the image pixels to the camera frame, each pixel location (u, v) is multiplied by the inverse of the intrinsic matrix, as shown in the following Equation:(2)$$\left[\begin{array}{c} X_{nd}\\ Y_{nd}\\ 1 \end{array}\right]=M_{I}^{-1}\left[\begin{array}{c} u\\ v\\ 1 \end{array}\right]$$

The resulting matrix is the same size as the image, where each pixel index (X_nd_, Y_nd_) contains normalized and distorted world-position values. To remove lens distortions, radial lens aberrations are corrected by dividing each pixel position with r = 1 + K_1_ × (X_nd_ + Y_nd_)^2^, where K_1_ is the radial aberration coefficient obtained from intrinsic parameters [23]. Note that experimental findings ignore tangential and high-degree radial distortion as they account for <0.1-pixel reprojection error. The distortion removal equation is
(3)$$\left[\begin{array}{c} X_{n}\\ Y_{n} \end{array}\right]\approx \frac{1}{1+k_{1}\left(x_{nd}^{2}+y_{nd}^{2}\right)}\left[\begin{array}{c} X_{nd}\\ Y_{nd} \end{array}\right],$$
where X_n_ and Y_n_ are the undistorted normalized positions. A scaling factor is applied to de-normalize the pixel positions to further transform pixels from the camera frame (X_n_, Y_n_) to world coordinates (X_w_, Y_w_). At X_w_ = 0 and Y_w_ = 0, the scaling factor $z_{c}\approx r_{3,3}*Z_{w}+t_{z}$, where $Z_{w}$ is known from the extrinsic parameter results. The relationship between the camera frame, $z_{c}$ scaling factor, and the world coordinate are expressed in the following Equation:(4)$$\left[\begin{array}{c} X_{n}\\ Y_{n}\\ 1 \end{array}\right]=\frac{1}{Z_{c}}\left[\begin{array}{c} X_{c}\\ Y_{c}\\ Z_{c} \end{array}\right]=\frac{1}{Z_{c}}\left(R\left[\begin{array}{c} X_{w}\\ Y_{w}\\ Z_{w} \end{array}\right]+T\right),$$
where R is the 3 × 3 rotation matrix of the extrinsic parameter, and T is the 3 × 1 translation matrix of the extrinsic parameters. Equation (4) is simplified to
(5)$$\left[\begin{array}{c} Z_{c}X_{n}-t_{x}-Z_{w}r_{1,3}\\ Z_{c}Y_{n}-t_{y}-Z_{w}r_{2,3} \end{array}\right]=\left[\begin{array}{cc} r_{1,1} & r_{1,2}\\ r_{2,1} & r_{2,2} \end{array}\right]\left[\begin{array}{c} X_{w}\\ Y_{w} \end{array}\right],$$
where $r_{m,n}$ are the components of the extrinsic rotation matrix; m stands for the matrix row, and n stands for the matrix column. Similarly, t_x_, t_y_, and t_z_ are the components of the extrinsic translation matrix. Therefore, X_w_ and Y_w_ is symbolically expressed in the following Equation:(6)$$\left[\begin{array}{c} X_{w}\\ Y_{w} \end{array}\right]={\left[\begin{array}{cc} r_{1,1} & r_{1,2}\\ r_{2,1} & r_{2,2} \end{array}\right]}^{-1}\left[\begin{array}{c} Z_{c}X_{n}-t_{x}-Z_{w}r_{1,3}\\ Z_{c}Y_{n}-t_{y}-Z_{w}r_{2,3} \end{array}\right]$$

Given that the above calculations transform pixel coordinates (u, v) to world coordinates (X_w_, Y_w_) in millimeters, measurements of laser shifts, expressed in millimeters, are integrated into a trigonometric model to reconstruct the Z_wo_ heights. The following section details the necessary trigonometric relations.

### 2.4. Laser-to-Camera Calibration Procedure

The laser calibration process is performed offline and consists of two types of scans. The first type records baseline recordings Y = Y_wb_ (for 250 images) by scanning the lasers across the FOV with no object. The second type of scan utilizes the same injection angles (τ) to capture images with shifted positions based on variable object heights (Z_wo_). A manufactured calibration phantom with known heights is used to facilitate this. The stage ranges from 5 to 50 mm with a step size of 5 mm, as depicted in Figure 6a. Subsequent scans are conducted while adjusting the phantom stage’s position within the FOV. Images from each scan with the laser scanning the calibration phantom data are kept for further calibration calculations. Essentially, this method produces 250 images representing laser shifts (Y = Y_wo_) associated with known Z_w_ and their corresponding injection angle (τ) that are used for later calculations. Figure 6b,c exemplify one of the baseline recordings alongside the corresponding calibration phantom image for red and green lasers.

Baseline projection positions (Y_wb_), object-shifted projection positions (Y_wo_), and their corresponding real-world heights (Z_wo_) are experimentally determined at specific galvanometer/mirror positions (τ). However, accurately measuring projected angles, θ, with the ground plane remains a challenge. Although Figure 7 and Appendix A detail the linear relationship between ∆τ and ∆θ where ∆θ = 2∆τ, the absolute θ values are unknown. The absolute θ values are contingent upon the galvanometer’s position in free space relative to the FOV. Each galvanometer requires independent calibration to discern its absolute θ values. Thus, a procedure to backward calculate θ values is formulated. This is accomplished by experimentally establishing the relationship between the projected shift positions (Y_wb_ − Y_wo_) for an object with a known height (Z_wo_) at a designated galvanometer motor position (τ). The laser projection angle θ is calculated using the following Equation:(7)$$\theta =\cot^{-1}\left(\frac{v_{b}-v_{o}}{Z_{wo}}\right),$$
which is a variation and simplification of Equation (1) under the experimental setting, $u_{b}$ = $u_{o}$ (only Y-shifts are considered).

Similar computational procedures are conducted for all disparity values (Y-shifts) to estimate the mapping between laser projection positions and the corresponding laser projection angles in the entire FOV. This mapping is described as {$v_{b}^{1}$, $v_{b}^{2}$, $v_{b}^{3}$, …, $v_{b}^{N}$ } → {θ^1^, θ^2^, θ^3^, …, θ^N^}, where N is the number of scan lines (N = 250 lines in our experiments). The mapping results are later utilized for online depth reconstruction. For a given laser projection position ($u_{b}^{i}$, $v_{b}^{i}$), based on the corresponding laser projection angle θ^i^ and laser position shift ($u_{b}^{o}$, $v_{b}^{o}$), the object height (Z_wo_) at position ($u_{o}^{i}$, $v_{o}^{i}$) can be estimated using the Equation:(8)$$Z_{wo}=\frac{\frac{r_{0,0}\;\times \;T_{z}}{R_{xy}^{-1}}\left(v_{b}-v_{o}\right)+\frac{{-r}_{1,0}\;\times \;T_{z}}{R_{xy}^{-1}}\;\left(u_{b}-u_{o}\right)}{\cot\left(\theta \right)+\frac{R_{xz}^{-1}}{R_{xy}^{-1}}+\left(\frac{-r_{1,0}\;\times \;r_{2,2}}{R_{xy}^{-1}}\right)\;u_{o}+\left(\frac{r_{0,0}\;\times \;r_{2,2}}{R_{xy}^{-1}}\right)v_{o}},$$
where the constants $R_{xy}^{-1}$ and $R_{xz}^{-1}$ are derived from the extrinsic matrix rotational components as follows:(9)$$R_{xy}^{-1}=r_{0,0}r_{1,1}-r_{0,1}\;r_{1,0},$$
(10)$$R_{xz}^{-1}=r_{1,0}r_{0,2}-r_{0,0}\;r_{1,2},$$
where $r_{m,n}$ are the components of the extrinsic matrixes, m stands for the matrix row, and n stands for the matrix column. Equation (8) integrates camera transformation results (Y_wb_ and Y_wo_) from Equation (6) and laser geometric transformations from Equation (1) for accurate depth reconstruction. This model is applied to every detected line laser pixel across the frame (column-wise) and is iteratively applied to each image in the scan sequence. Comprehensive depth reconstruction is achieved by linearly interpolating missing gaps in the depth information not captured in the scanned images.

### 2.5. Depth Resolution and Maximum Depth Estimates

For scanning laser triangulation, theoretical depth resolution is expressed as the ratio of the acquired depth estimate to the laser shift in the Y-direction. Equation (11) is a variation of Equation (8), presenting the calculation of the system’s theoretical resolution by simplifying the X-axis components as follows:(11)$$Depth\;Resolution\left|\frac{z_{wo}}{v_{b}-v_{o}}\right|\approx \;\left|\frac{\frac{r_{0,0}\;\times \;T_{z}}{R_{xy}^{-1}}}{\cot\theta +\left(\frac{-r_{1,0}\;\times \;r_{2,2}}{R_{xy}^{-1}}\right)+\left(\frac{r_{0,0}\;\times \;r_{2,2}}{R_{xy}^{-1}}\right)v_{o}\;}\right|$$

When expressed for a single line laser, the resolution is strongly influenced by the θ angle in the denominator. A reduced θ inherently yields a finer depth resolution. However, this concurrently restricts the maximal height measurement, given that a theoretical maximum height is defined by $Z_{w}^{max}=∆Y^{max}*\;\tan\left(\theta \right)$. Therefore, a tradeoff exists between the maximum height measurement and depth resolution. In the experimental setup, this balance is acknowledged by strategically positioning the galvanometers relative to the FOV, where most θ values are close to 45 degrees. Additionally, a more accurate, experimentally derived resolution accounts for depth values from both lasers, averaging their readings based on each laser’s θ value. Since the lasers scan from opposite directions, a high θ value in one laser ensures the other possesses a lower θ value. This is an intrinsic advantage of our system geometric setup, which contains scanning line lasers from opposite directions.

### 2.6. Performance Assessment and Agricultural Use Cases

Multiple manufactured calibration stages are scanned and reconstructed to assess the developed depth imaging system and gauge the setup’s accuracy and depth resolution. These stages are an unbiased benchmark considering their known and precisely measured heights. The first stage features a sloping periphery that enables us to quantitively examine the system performance in accurately and precisely reconstructing sloped surfaces. The second stage comprises a 3D-printed pyramid with a 0.1 mm layer resolution. It is designed to be symmetrical and has an apex, so it is positioned at the center of the FOV for a robust assessment. This arrangement enables assessing the system’s performance across varying projection angles θ and different object heights along the X_w_ and Y_w_ axes.

Finally, to demonstrate the capabilities and versatility of the depth imaging system, two types of agricultural products are scanned: Chesapeake Blue crabs and White Button mushrooms. Crabs’ medium size and complex morphology highlight the system’s ability to capture depth maps of intricate and overlapping objects in a pile. On the other hand, mushrooms, which are smaller and feature a uniform texture and color, present a different challenge due to their tendency to cluster tightly in piles, making individual heights difficult to distinguish.

To illustrate the advantages of the dual line-laser system and its geometric setup, the process is initiated by capturing colored images. Subsequently, the scanning routine acquires, processes, and reconstructs depth maps. To underscore the utility of the dual-laser setup, a depth map from each laser is generated independently before merging the two to reveal their synergistic impact. Nevertheless, the system operates concurrently using both lasers in real-time in normal operations.

## 3. Results

Figure 8 displays the experimental setup for the dual line-laser active system, featuring a versatile aluminum T-slotted frame for adaptable component placement. The camera is mounted 1000 mm above the tabletop. The beige HDPE tabletop is ideal for reflecting both colors without being diffusive. The experimental setup accommodates a range of static piles of agricultural products without any conveyor movement. The setup is mobile and can be aligned alongside robots or other processing machines.

### 3.1. Laser Calibration Analysis

Figure 9 depicts experimental results for the laser calibration step. The plots illustrate the relationship between laser projection angles (θ) and the corresponding laser-projected positions (v). This relationship exhibits a high degree of linearity, with an R^2^ = 0.998, which is consistent with our expectations given the linear voltage-controlling strategy that was previously mentioned. The lasers begin scanning from the opposite sides of the FOV, leading to a positive galvanometer offset for the red laser and a negative one for the green laser.

In the scanning experiments, the total input voltage change for the galvanometer is 4.55 V. The galvanometer’s mechanical position scale is 0.5 V/degree, translating to a ∆τ = 9.1 degrees. Following the linear relationship outlined in Appendix A, the theoretical calculation of ∆θ = 2∆τ should yield an 18.2-degree change in the laser projection angle. This corroborates the experimental findings for both lasers; specifically, with a linear correlation slope of 0.018 over a range of 1030 pixels, the experimental ranges obtained are ∆θ_red_ = 18.54 degrees and ∆θ_green_ = 18.54 degrees. The discrepancies between the experimental and theoretical results are within the expected 0.4-degree resolution of the ∆θ (0.2 degrees galvanometer resolution multiplied by two as per ∆θ = 2∆τ). With the laser mapping equation in Figure 9, objects’ heights are reconstructed using Equation (8).

### 3.2. Maximum Depth Estimates

The theoretical maximum height $Z_{w}^{max}=∆Y^{max}*\;\tan\left(\theta \right)$ when θ = θ_0_. In the experimental setup, the FOV has a ${\Delta Y}^{max}=350\;mm$ at θ_red_ = 51.741 degrees and θ_green_ = 50.975 degrees obtained from Figure 5. Thus, $Z_{w}^{max}\left(red\right)=443.831\;mm$ and $Z_{w}^{max}\left(green\right)=431.828\;mm$. The maximum height is only effective on the lateral sections (vertically) of the FOV. In the middle of the FOV, the maximum height is lower because $Z_{w}^{max}=\frac{\Delta Y(max)}{2}*\;\tan\left(\theta \right)$ at θ = θ_125_. Effectively, in the middle of the FOV, $Z_{w}^{max}\left(red\right)=306.124\;mm$ and $Z_{w}^{max}\left(green\right)=315.851\;mm$. The maximum heights between the $Z_{w}^{max}$ at θ = θ_0_ and θ_125_ can be linearly interpolated.

### 3.3. Depth Imaging Outcomes of Known Heights

The calibration phantom in Figure 10 is reconstructed to validate the system’s efficacy for sloped surfaces. The stage has a flat top with a 50 mm height and 45-degree sloped sides. Figure 10 presents the corresponding cross-sectional height estimation of the phantom. The two slopes of the phantom were successfully reconstructed with an R^2^ value of 0.9996. The height estimation error of the flat stage is 0.57 mm, and the standard derivation is 0.15 mm.

For the 3D-printed pyramid-shaped stage, the acquired color image, depth reconstruction for each laser, and their final merged result are shown in Figure 11. Depth maps reconstructed independently by each laser show gaps in the pyramid due to obstruction. These gaps are filled in the merged depth map, showing the synergistic effect of the two lasers. Masks are used to analyze the reconstructed heights compared to ground truth, and their summarized results are shown in Figure 11 and detailed in Table 1. The overall system has a Mean Squared Error (MSE) of 0.3 mm and a standard deviation (STD) of 0.5 mm. The number of pixels tested for these results is also indicated for each height level.

### 3.4. Piled Object–Depth Maps from Dual Active Laser Imaging

The results of the depth reconstruction of the Chesapeake Blue crab and White Button mushroom are shown in Figure 12. The green laser paths are obstructed from illuminating the upper margin of objects in the FOV. Likewise, the red laser paths are obstructed from illuminating the bottom margin of objects in the FOV. However, when both lasers’ contributions are overlaid, the resulting full-depth map shows the lasers’ synergistic effects.

In contrast to previous analyses of objects with known heights, it is challenging to determine ground truth data for a random pile of agricultural commodities. The benefits of including a second scanning laser are quantitatively scrutinized by reporting each laser’s contribution to the depth map. In the crab case, the red laser independently contributed 80% of the depth map, missing the rest due to obstructions. The green laser independently contributed 79%. Of these red and green depth map pixels, both lasers overlapped by 65%. Synergistically, over 95% of all pixels were filled with depth data in the full reconstruction.

Similarly, in the case of the mushrooms, the red laser independently contributed 77%, and the green laser contributed 72%. Of these pixels, both lasers overlapped in 56% of the pixels. In the full reconstruction, 93% of all pixels were filled with depth data, and the rest were data not detected by either laser. These results ultimately highlight the importance of the geometric configuration in filling the missing gaps in piled products.

### 3.5. Image Acquisition and Depth Reconstruction Duration

Timing analysis is crucial for the system’s applicability to industrial settings. On average, capturing a colored background image followed by 250 scanning images requires approximately 740 ms, translating to an effective frame rate of about 338 Frames Per Second (FPS). The acquisition speed is constrained by GenICam protocol and PCI-e bandwidth limitations. Subsequent image processing tasks—such as image rotation, background subtraction, and laser color segmentation—consume an additional 770 ms. Laser position detection requires 1130 ms, while the trigonometric calculations essential for the full-depth reconstruction algorithms are completed in 320 ms. Processing times are obtained using the Intel Core i9-12900KF processor (Intel Corporation, Santa Clara, CA, USA). The time needed for image acquisition and depth reconstruction is approximately 1.8 s. The system stays static for only the acquisition time (740 ms).

## 4. Discussion

The depth imaging system has numerous advantages and far-reaching implications for industrial applications. As of the writing of this study, the system can be constructed for less than USD 5000, making it cheap and accessible for implementers in industrial plants. Unlike traditional line scan cameras that require conveyor motion, the active scanning approach captures the depth map of static items. This feature is particularly beneficial for imaging piled products, where conveyor movement would disrupt their arrangement and registered locations. The system enables the acquisition of colored images with an overlaid depth map through a single camera. This is in contrast with approaches like TOF and 2D structured light, which often require separate color and depth imaging cameras and an additional registration step needed to match color and depth images. The dual active laser imaging minimizes the risk of misalignment between depth and color data and avoids extra registration operations. Such a feature proves particularly beneficial for industrial lines requiring RGB-D data for tasks like sorting or visual servoing [1]. Finally, in terms of robustness, the system not only rivals high-end, industrial-grade depth cameras but also adapts effectively to varying ambient light conditions. Camera Link’s Low-Voltage Differential Signaling (LVDS) communication protocols, which resist electromagnetic interference from nearby industrial actuators, are incorporated to enhance reliability.

Important insights and capabilities of the system are uncovered through experimentation. As the results show, the laser redundancy enables the illumination of crevices and areas obstructed by nearby apices. Laser redundancy captures height information at the image’s top and bottom boundaries, areas often neglected due to laser shifts falling outside the FOV. The only limitation arises when one laser is obstructed and the corresponding laser shift from the second also falls outside the FOV. The proposed setup accounts for uneven laser thickness across the FOV due to changing incident angles and laser defocusing at various optical paths. A uniform depth map is reconstructed by averaging data from both depth maps. In the 3D-printed pyramid evaluation, the tested pixels obtained consistently had a lower estimation than ground truth values. Although the errors are in the order of the submillimeter, it is possibly due to the lens defocusing as items become closer to the camera, which is a possible limitation of the system.

A thinner line laser and a larger number of laser illuminations (N values) lead to higher depth map lateral resolution, whereas smaller N values save image acquisition time. Therefore, active scanning systems have an intrinsic tradeoff between image lateral resolution and scanning time. The reconstruction performance is determined by the projected laser angle θs for depth map accuracy, as shown in Figure 11. A smaller θ leads to better depth resolution because it has a larger Y shift; however, it would inadvertently limit the maximum height measurement. Given a laser angle projection θ and the projection position Y_wb_, the theoretical maximum height is Y_wb_ × tan(θ). Hence, a tradeoff exists between maximum height measurement and depth resolution. In the proposed system, the galvanometer units were strategically placed close to 45 degrees to balance the depth accuracy and maximum height. The plane-constrained configuration measures object heights from Z = 0, where the world Z-axis points upward (towards the camera). Therefore, the maximum depth estimates reported for the proposed system are synonymous with the minimum working distance reported by other depth imaging methods.

The experimental results show the efficacy of the dual active line-laser scanning strategy and its advantages in reconstructing the depth imaging of piled products. It is essential to place the results within the broader context of current research. A recent study by Xiang and Wang [24] provided a comparative analysis of depth imaging techniques in food and agriculture applications assessing depth imaging approaches, use cases, advantages, disadvantages, and price ranges. Besides high-depth reconstruction performance, the dual line laser approach presents a cost-effective implementation and a robust calibration process. This advances the state-of-the-art for laser scanning relative to other modalities. Previous studies, such as those by Schlarp et al. and Yang et al., reported higher resolution systems than the current study but were limited by a working distance of approximately 100 mm and small FOVs [10,25,26,27,28,29]. Small working distance inevitably leads to a small height range of the depth map, which is critical in imaging piles. While these studies present significant advancements in modeling, calibration, and depth reconstruction using a single galvanometric line laser, the proposed dual line-laser system distinguishes itself by imaging a significantly larger FOV with a long working distance and a more extended height range of the depth map while maintaining submillimeter MSE and STD. Other research efforts [30,31] attempted to develop single galvanometric scanning for longer working distances of 1 to 2 m, but they reported higher margins of error that are as high as 10 mm. These findings underscore the advantages of utilizing a dual line laser approach and merging their respective depth maps to mitigate the impact of outliers from a single depth map.

The proposed approach is designed for flexibility and customization, enabling developers to optimize hardware according to required scanning speeds and desired accuracy. The system’s speed, crucial for higher throughput in industrial settings, hinges on three key aspects: scanning speed, communication, and image processing. The scanning hardware triggering routine minimizes frame-to-frame crosstalk while maintaining high scanning rates. Additionally, line laser utilization yields faster scanning compared to dot scanners. Future work should explore unidirectional polygonal mirrors to maneuver the light path faster than single-axis galvanometers used in experiments. For image acquisition, high-speed industrial-grade CMOS cameras reduce scan durations. These cameras offer dynamic shutter speed control—higher exposure times for well-lit colored frames and lower exposure for increased FPS while effectively reducing ambient light interference. Future upgrades will incorporate advanced Camera Link communication speed up, utilizing fiber optics for enhanced bandwidth/frame rate support. Lastly, porting the algorithm and calibration data to an onboard FPGA enables immediate depth map generation, significantly accelerating processing [32] compared to computer processors that incur overhead.

The limitations of the dual active line-laser scanning strategy in industrial applications are multifaceted and contain inherent tradeoffs. First, the larger the FOV, the lower the depth map lateral resolution at the same line laser width and scanning parameters. Second, external noise influencing laser positioning in the image frame degrades depth reconstruction and the lateral resolution of depth maps. Among the known contributing noise variables are laser thickness, hardware errors associated with the galvanometer, and post-processing algorithms that are crucial in determining the centroid of the laser position. Third, the system requires precise calibration and cannot obtain reliable measurements if the camera settings are altered. Fourth, as is common with most laser depth scanners, scanning products with reflective surfaces is usually challenging due to specular reflection. A blue laser can partially mitigate this problem; however, selecting a camera with optimal sensitivity to the blue light spectrum is essential. The colors of the objects imaged also play a significant role because the light reflected off objects’ surfaces determines the RGB threshold values for laser detection and localization. Objects with darker surfaces absorb visible light, but higher laser powers can be utilized to mitigate color absorption. Finally, high laser power poses eye safety concerns. Protective eyewear must accommodate laser frequency, optical density, and diameter in industrial workspaces.

## 5. Conclusions

An active dual line-laser scanning system was introduced. The system employed two line lasers coupled with programmable galvanometers to scan the FOV. Once calibrated to real-world heights, the system produced depth maps with exceptional performance, achieving an MSE of 0.3 mm and an STD of 0.5 mm. The method has advantages over consumer-grade depth cameras that rely on 2D structured light and Time-of-Flight (TOF) methods, which show standard deviations ranging from 1 to 5 mm at a 1m distance. Leveraging a unique geometrical configuration and laser redundancy, the system resolves depth in challenging environments where overhead cameras face obstructions posed by piles and concavities. The scan results of Chesapeake Blue crabs and White Button mushrooms showed the synergistic effects of the two lasers in illuminating occluded areas. Although initially designed for agricultural applications, the system is versatile enough to be adapted for a wide array of textured and textureless products such as medical equipment and automobile parts. Future work will enhance this design with computer vision algorithms such as 3D object segmentation and robotics to transform unordered piles of products into isolated items for streamlined food processing.

## 6. Patents

U.S. Patent Pending: Tao, Y., D. Wang, and M. Ali. 2022. Systems and Methods for Machine Vision Robotic Processing. Machine Vision Guided Robotic Loading and Processing on Manufacturing Lines. U.S. Patent Application No. 17/963,156.

## Figures and Tables

**Figure 1 sensors-24-02385-f001:**
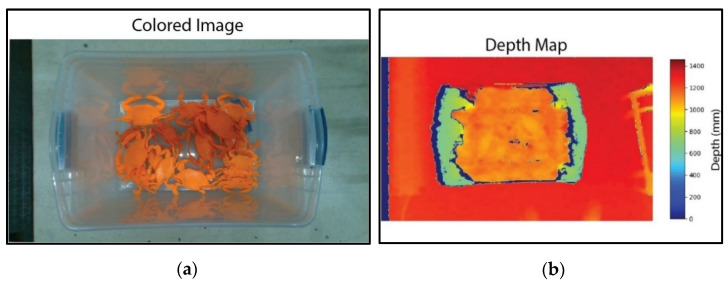
Commercial RGBD cameras have poor depth resolution. (**a**) RGB image and (**b**) depth map acquisition using Intel Real Sense L515. Notably, the low depth resolution cannot distinguish crabs in a bucket at ~1 m image acquisition distance. The color camera and depth map have different fields of view due to different sensor sizes.

**Figure 2 sensors-24-02385-f002:**
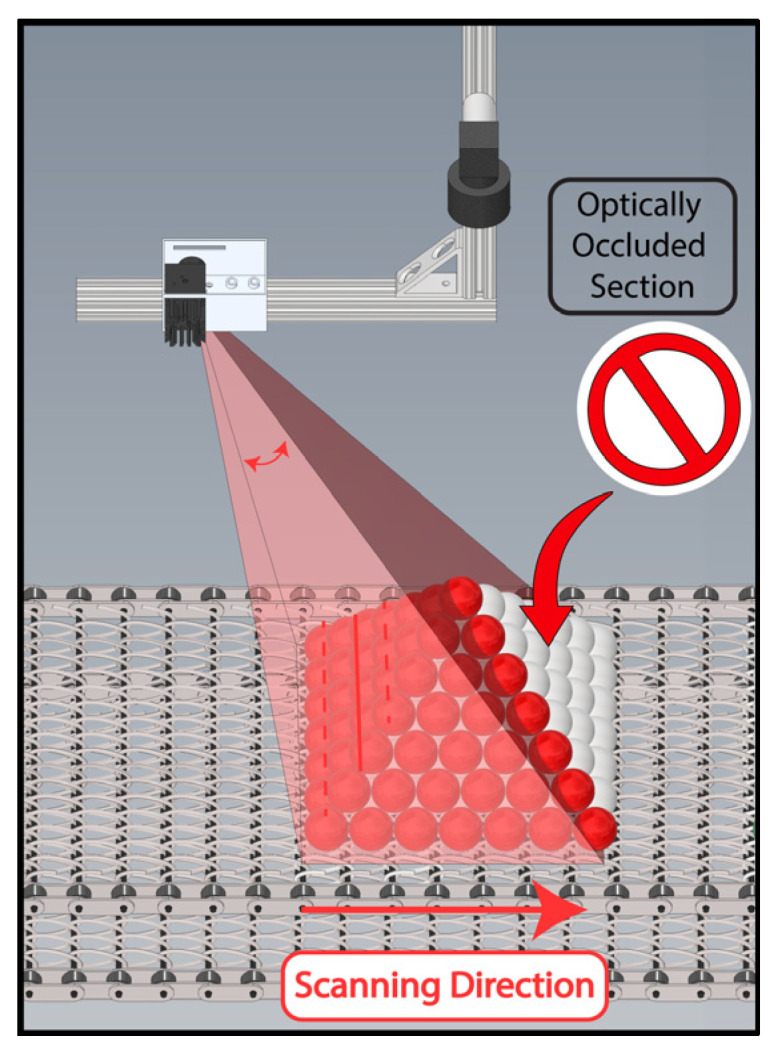
Limitations of line-laser triangulation, whether through conveyor motion or active line scanning, as it only illuminates and reconstructs a portion of a pile. The method fails to reach the far side, restricted by the inherent constraints of the laser’s optical path.

**Figure 3 sensors-24-02385-f003:**
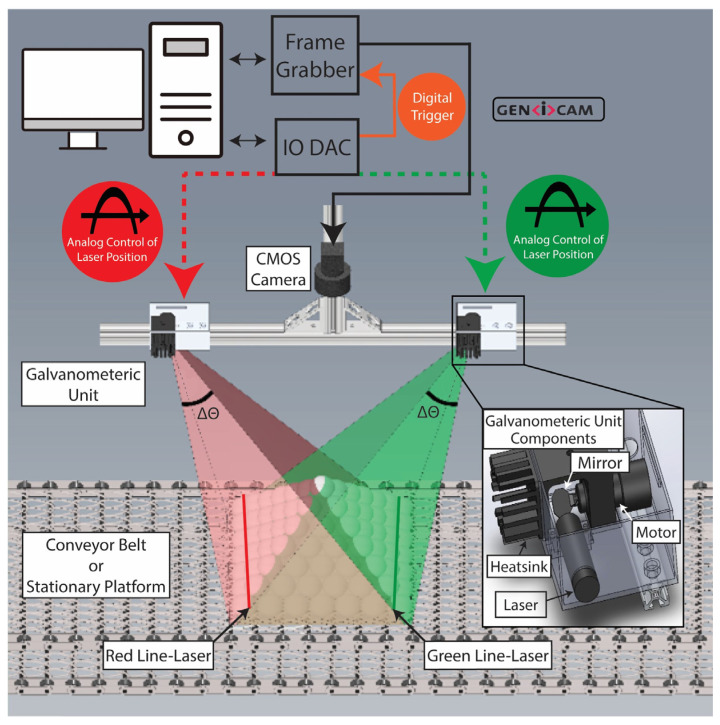
Hardware flowchart: centralized computer triggers image acquisition through a frame grabber and laser control via IO DAC. The overhead CMOS camera is positioned between two galvanometric units. Each unit consists of red or green line lasers, a galvanometer (motor), a mirror, and a heatsink. The red and green laser cones represent the laser optical reach.

**Figure 4 sensors-24-02385-f004:**
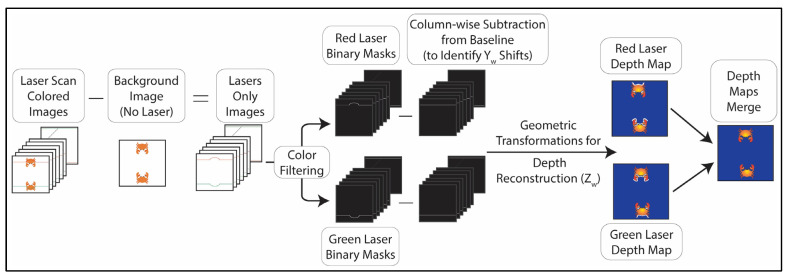
The post-processing procedure starts with laser-image subtraction from the background image to isolate the laser signature from object features. The laser-only images undergo color threshold to identify laser positions. The Y-shifts are determined by performing column-wise subtraction between laser baseline position and object-based position. These Y-shifts are converted into depth values through laser triangulation’s trigonometric calculations. Finally, both depth reconstructions are combined to obtain a comprehensive depth map.

**Figure 5 sensors-24-02385-f005:**
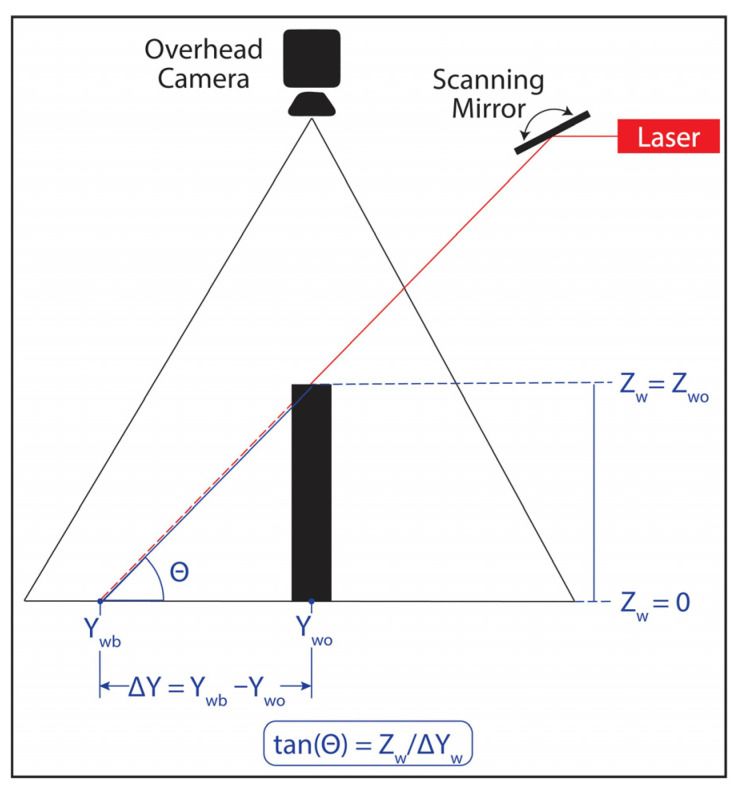
Simplified diagram of optical triangulation trigonometry. Y_wb_ is the Y-axis baseline position, Y_wo_ is the location of the laser shift, Z_w_ is the object’s height in mm, and θ is the projected angle with the ground (Z_w_ = 0).

**Figure 6 sensors-24-02385-f006:**
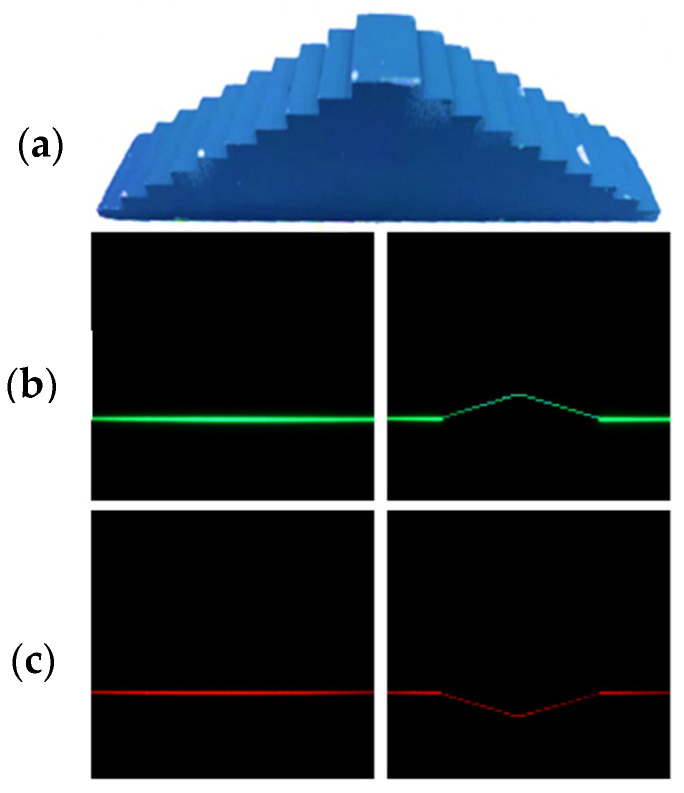
(**a**) Calibration phantom with 5 to 50 mm steps in 5 mm increments for depth calibration. (**b**) Green baseline and shifted laser recordings at a specific galvanometer position (τ). (**c**) Comparable red line laser images. Each set of laser images represents one of 250 images captured during the laser calibration process.

**Figure 7 sensors-24-02385-f007:**
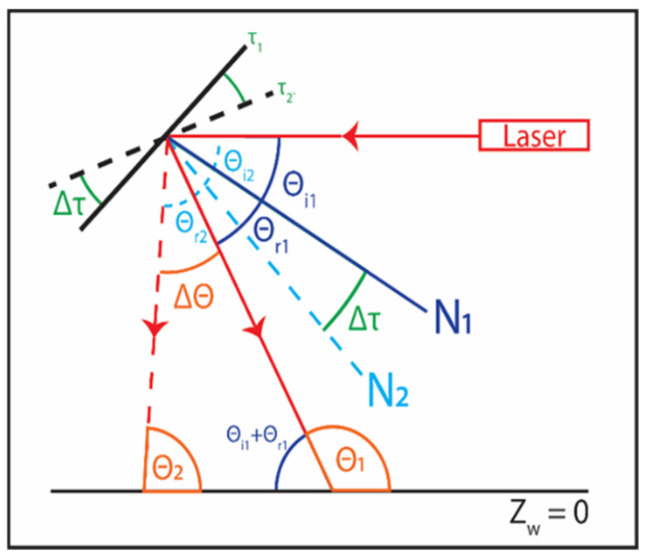
Diagram illustrating the geometric relationships between the galvanometer’s mirror position and the resulting angles’ θs with the ground, where the ground is defined as Z_w_ = 0. Here, N represents the normal line from the mirror’s positions, τ denotes the angle position of the mirror, θ_i_ is the incident angle, and θ_r_ is the reflected angle. By applying the law of reflection and considering corresponding angles, the change in angle is given by ∆θ = 2∆τ.

**Figure 8 sensors-24-02385-f008:**
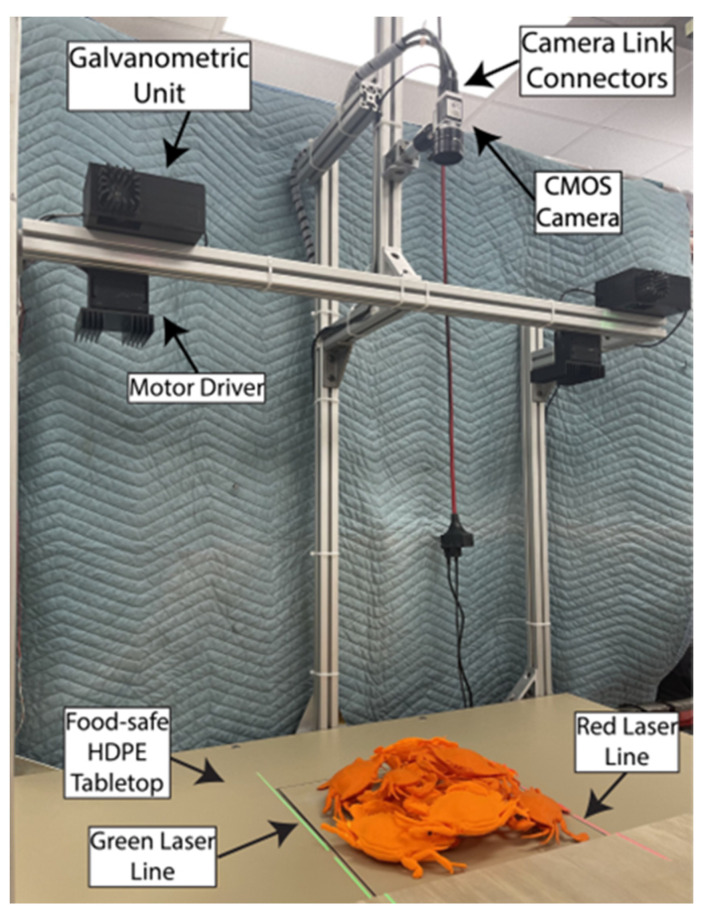
Experimental setup of the dual line-laser active scanning system.

**Figure 9 sensors-24-02385-f009:**
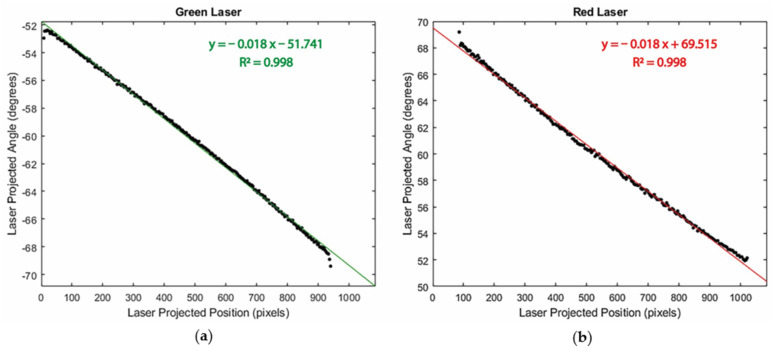
Relationship between laser-projected angle and laser position for (**a**) green laser and (**b**) red laser. Lasers have opposing angle offsets due to their geometrical configuration at opposite ends of the field of view.

**Figure 10 sensors-24-02385-f010:**
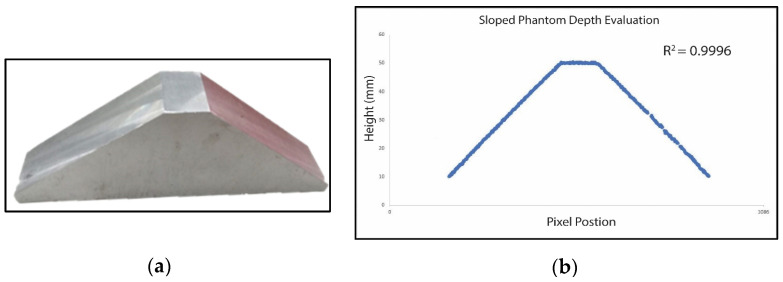
(**a**) Sloped stainless steel phantom to examine reconstruction of sloped objects; (**b**) height estimation across 1086 pixels. The R^2^ value for the reconstructed slope is 0.9996.

**Figure 11 sensors-24-02385-f011:**
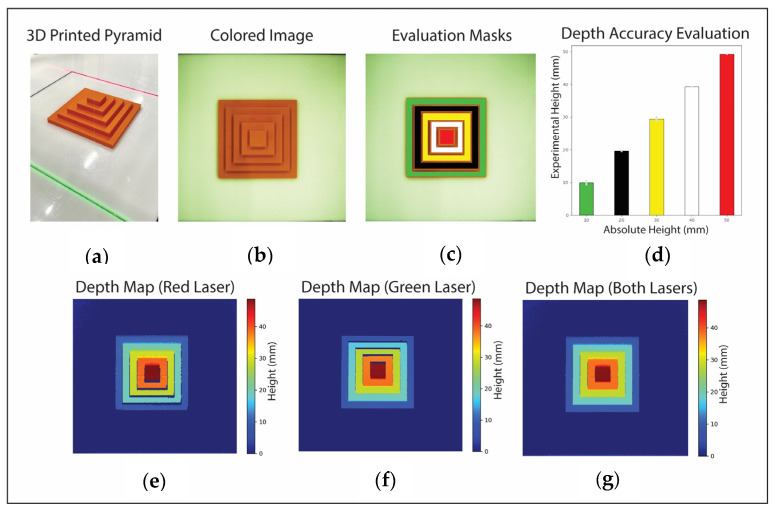
Depth reconstruction outcomes: (**a**) a 3D-printed pyramid as the object of study; (**b**) the colored image captured by the overhead camera; (**c**) masks applied for depth data assessment; (**d**) the depth results with standard deviation for each level. Depth maps using the (**e**) red laser and (**f**) green laser, each reconstructed independently; (**g**) the final depth reconstruction of both lasers merged.

**Figure 12 sensors-24-02385-f012:**
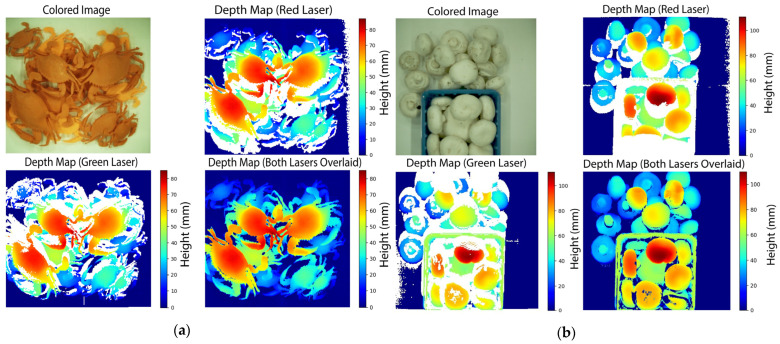
(**a**) Three-dimensional-printed Chesapeake Blue crab-colored image, depth reconstruction using red and green lasers independently, and the merged depth map of both lasers. Similarly, (**b**) mushroom-colored image, depth reconstruction using red and green laser independently, and the merged depth map of both lasers. The final depth map showcases the synergistic effects of both lasers’ contributions.

**Table 1 sensors-24-02385-t001:** Three-dimensional-printed pyramid depth reconstruction results. The ground truth heights are the heights at which the pyramid is 3D printed. Mean heights are the average of all values obtained at the location of the ground truth masks. Mean Squared Error (MSE), standard deviation (STD), and pixels tested showcase the system performance and the number of pixels for each ground truth height of the pyramid.

Ground Truth Heights (mm)	Mean Heights (mm)	STD (mm)	MSE (mm)	Pixels Tested
50	49.161	0.198	0.702	6639
40	39.216	0.259	0.6143	17,605
30	29.392	0.873	0.368	34,591
20	19.597	0.350	0.161	47,834
10	9.938	0.861	0.003	58,821

## Data Availability

The raw data supporting the conclusions of this article will be made available by the authors upon request.

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
