# Peer review of "Active Dual Line-Laser Scanning for Depth Imaging of Piled Agricultural Commodities for Itemized Processing Lines"

_sensors, 2024, doi:10.3390/s24082385_

Round 1

Reviewer 1 Report

Comments and Suggestions for Authors

The author has achieved precise depth imaging of stacked products through structured methods like triangulation. The dual-line laser scanning approach, combined with system calibration, ensures that most pixels are filled with depth data. This advancement benefits robotic perception for individual item retrieval from a stack of food, enabling sequential food processing. However, the reviewer has several questions of interest:
1.    What is the cost of this system, and has it been applied in industrial production with satisfactory performance?
2.    The paper mentions that TOF-based sensors have low depth reconstruction accuracy, but to my knowledge, single-photon lidar based on DTOF can achieve millimeter-level depth resolution.
3.    The paper mentions the limited use of conveyor motion for target 3D imaging and the use of a scanning mirror, which seems like an obvious solution. What are the challenges in implementing this scanning mirror? Is it primarily related to system calibration?

Reviewer 2 Report

Comments and Suggestions for Authors

In this manuscript, the authors developed an active dual line-laser scanning system for depth imaging of stationary piled items, which was demonstrated to effectively overcome occlusions while achieving a large FOV from a long working distance with decent imaging resolution and reasonable image acquisition and processing time.  This manuscript overall is well-written, and demonstrated a new depth imaging system that provides a relatively straightforward yet effective solution to the applications in itemized processing lines for piled agricultural produce. Would recommend accepting this paper after the following questions are addressed.

1.      Please double check eq. 7 since it seems to be in conflict with eq.1. Is the cot^(-1)  meant to be cot?

2.      In section 2.3, eq. 4-6, the variables R and T and their components such as r11and r13  are not defined which makes it difficult to follow the math. Please include definitions for those variables.

3.     Eq.(11) implies that when there is no x direction shift(ub-u0à0), the depth resolution would be infinitely small, which contradicts previous discussions and the physical intuition. Also, please add the derivation from eq(8) to eq(11) or at least clarify what approximations/assumptions are made here.

4.      The authors discussed about the trade-offs between the height-resolution and maximum height. Could the authors provide a theoretical and/or experimental maximum height of the demonstrated system and compare it with the other systems?

Reviewer 3 Report

Comments and Suggestions for Authors

The present manuscript is really intriguing and introduces an active dual line-laser scanning system that obviates the necessity for conveyor motion in order to produce depth maps when imaging static piled objects, such as a pile of crabs on a table. With the few minor revisions delineated below, the manuscript is suitable for publication in the Sensors journal.

Line 153 specifies that the thickness of the laser beam is 1.4 mm. The text does not refer to the beam divergence factor. Does this thickness pertain to the point at which the laser beam initiates from the sensor or the point at which it makes contact with the object? Acquiring such measurements (MSE of 0.3 mm and an STD of 0.5 mm) from a 1.4 mm sensor is possible for solid objects like the pyramid. However, it would be challenging to achieve these measurements for the smaller test objects like the crabs and the mushrooms.

Figure 2 depicts a laser scanner positioned at an oblique angle, resulting in noticeable occlusions on the object (pyramid). This claim is utilized to establish the originality of the dual line-laser scanning system put forth in this manuscript. What would be the distinction between the suggested dual line-laser configuration and a single line-laser scanner configuration in nadir angle, considering that both setups produce depth maps from a nadir perspective? It is suggested that the authors should change accordingly the introduction and the conclusion sections.

Lastly, it is recommended to discuss the scenario of a scanning system that utilizes a collection of camera-rigs. These camera-rigs are commonly employed in various fields such as culture and human 3D reconstruction. The scanning system operates in a controlled lighting environment and is significantly more cost-effective. The photography process is completed in a matter of a second, but the generation of depth maps necessitates a relatively lengthy and computationally intensive photogrammetric processing.
